# Refined PHD Filter for Multi-Target Tracking under Low Detection Probability

**DOI:** 10.3390/s19132842

**Published:** 2019-06-26

**Authors:** Sen Wang, Qinglong Bao, Zengping Chen

**Affiliations:** 1National Key Laboratory of Science and Technology on ATR, National University of Defense Technology, Changsha 410073, China; 2School of Electronics and Communication Engineering, SUN YAT-SEN University, Guangzhou 510275, China

**Keywords:** refined PHD filter, low detection probability, continuous miss detection, radar multi-target tracking, survival probability, target labels, posterior weight revision, sequential probability ratio test, hypothesis test

## Abstract

Radar target detection probability will decrease as the target echo signal-to-noise ratio (SNR) decreases, which has an adverse influence on the result of multi-target tracking. The performances of standard multi-target tracking algorithms degrade significantly under low detection probability in practice, especially when continuous miss detection occurs. Based on sequential Monte Carlo implementation of Probability Hypothesis Density (PHD) filter, this paper proposes a heuristic method called the Refined PHD (R-PHD) filter to improve multi-target tracking performance under low detection probability. In detail, this paper defines a survival probability which is dependent on target state, and labels individual extracted targets and corresponding particles. When miss detection occurs due to low detection probability, posterior particle weights will be revised according to the prediction step. Finally, we transform the target confirmation problem into a hypothesis test problem, and utilize sequential probability ratio test to distinguish real targets and false alarms in real time. Computer simulations with respect to different detection probabilities, average numbers of false alarms and continuous miss detection durations are provided to corroborate the superiority of the proposed method, compared with standard PHD filter, Cardinalized PHD (CPHD) filter and Cardinality Balanced Multi-target Multi-Bernoulli (CBMeMBer) filter.

## 1. Introduction

The objective of Multi-Target Tracking (MTT) is to jointly estimate the number of targets and their individual states, and to provide target tracks or trajectories, from a sequence of measurements provided by sensing devices such as radar [1], sonar [2], or cameras [3]. Traditional MTT algorithms, including Joint Probabilistic Data Association Filter (JPDAF) [4] and Multiple Hypothesis Tracking (MHT) [5], always transform the multi-target tracking problem into multiple independent single-target tracking problems by data association processing according to a certain distance criterion. Association error resulting from complex scene deteriorates the tracking performance of JPDAF and MHT.

In recent years, multi-source multi-target information fusion theory based on Random Finite Sets (RFS) provides a unified and scientific mathematical basis for multi-sensor multi-target tracking problem [6,7]. Different from traditional heuristic methods, multi-target tracking methods based on RFS strictly describe target birth, death, spawning, miss detection and clutters in multi-target tracking process, directly estimate number and state of targets, and even provide target tracks or trajectories by modeling multi-target states and sensor measurements as RFS or labeled RFS, which has the best performance in Bayesian sense.

The multi-target Bayes filter is difficult to implement. Fortunately, some advanced approximations have been proposed, such as the Probability Hypothesis Density (PHD) filter [8,9,10], the Cardinalized PHD (CPHD) filter [11,12], the Multi-target Multi-Bernoulli (MeMBer) filter [13] and the Cardinality Balanced MeMBer (CBMeMBer) filter [14]. More recently, multi-target tracking algorithms based on labeled random finite sets have been proposed [15,16,17,18,19,20,21,22,23], and it can obtain track-valued estimates of individual targets without the need for post-processing, such as the Generalized Labeled Multi-Bernoulli (GLMB) filter [15] and Labeled Multi-Bernoulli (LMB) filter [16].

Since Sequential Monte Carlo (SMC) implementation and Gaussian Mixture (GM) implementation of PHD filter were proposed, the PHD filter has attracted significant attention in multi-target tracking research. To reduce the computational complexity of the PHD filter, several gating strategies were introduced to exclude clutter observation participating in filter updating [24,25]. To obtain target states from posterior PHD, several multi-target state extraction algorithms have been proposed, such as clustering [26,27] and data-driven methods [28,29,30,31]. To fuse information from multiple observation system, multi-sensor multi-target tracking filters based on PHD were proposed [32,33,34]. To track maneuvering targets, traditional multi-model method was introduced to PHD filter [35]. Faced with unknown backgrounds, such as unknown detection probability, unknown clutter parameter, several improved PHD filters can estimate background parameters while tracking [36,37]. In non-standard target observation model, several improved PHD filters were proposed to track extended target [38,39].

The standard PHD filter has considered the influence of the detection probability on multi-target tracking, but its performance degrades significantly under low detection probability in practice, especially when continuous miss detection occurs. For example, the posterior particle weights of a SMC-PHD filter will become small under continuous miss detection, and corresponding particles may be eliminated from the particle pool and then the undetected target will be lost.

Several recent works have made some attempts [40,41]. Based on the GM-PHD filter, the Refined GM-PHD (RGM-PHD) filter [40] was proposed to improve the performance of the GM-PHD filter under continuous miss detection. This method is effective in terms of various detection probabilities, false alarm rates and continuous miss detection rates. However, some key parameters of the RGM-PHD filter, including the penalty coefficient and the reward coefficient, are determined without explicit formula, which is difficult to be generalized to other applications. Also based on GM implementation of PHD filter, a novel target state estimate method was integrated into three improved GM-PHD filters [41], which results in better tracking performance in imperfect detection probability scenarios. However, lower bound of detection probability in simulations is set as 0.8, which can’t sufficiently illustrate the effectiveness of the method under low detection probability.

In this paper, based on SMC implementation of PHD filter, we propose a heuristic method called Refined PHD (R-PHD) filter to improve multi-target tracking performance under low detection probability. First, survival probability dependent on target state is defined, which is based on the hypothesis that target enter and exit sensor Fields of View (FoV) usually occur at the boundary. Then, individual target and particle are assigned a unique label, which is utilized to confirm if miss detection occurs for each target and identify particles representing the undetected target. When miss detection occurs, posterior weights will be revised according to the prediction step. The key of the proposed method is to distinguish real targets and false alarms. This paper binarizes the likelihood function of individual extracted target, which is approximated as a random variable obeying two-point distribution. When extracted target is a real target, success probability of the two-point distribution is approximatively the detection probability. When extracted target is a false alarm, the success probability is approximatively a very small value. Then this paper transforms target confirmation problem into a hypothesis test problem, and utilizes Sequential Probability Ratio Test (SPRT) [42] to confirm real targets in real time. After target extraction at each time, we mark each extracted target as a real target or a false alarm, or make no decision according to test statistic.

The rest of the paper is organized as follows. Section 2 reviews probability hypothesis density filter and corresponding SMC implementation. Section 3 proposes the refined PHD filter in detail. Computer simulations illustrating the effectiveness and the performance of the proposed method are provided in Section 4. Finally, Section 5 presents the conclusion.

## 2. Background

This section will introduce the probability hypothesis density filter. Furthermore, SMC implementation of the PHD filter will also be reviewed.

### 2.1. PHD Filter

The probability hypothesis density is defined as the first-order statistical moment of multiple-target posterior distribution. Similar to the constant-gain Kalman filter in single-target filtering, the PHD filter is the first-order moment approximation of the multi-target Bayes filter, which only recursively propagates first-order multi-target moments by time prediction and data-update steps. Suppose $D_{k-1|k-1}(x)$ is the PHD at time k−1, the predictor equation of the PHD filter can be expressed as
(1)$$D_{k|k-1}(x)=b_{k|k-1}(x)+\int \left[p_{S}(x^{\prime })\cdot f_{k|k-1}(x|x^{\prime })+b_{k|k-1}(x|x^{\prime })\right]\cdot D_{k-1|k-1}(x^{\prime })dx^{\prime },$$
where $f_{k|k-1}(x|x^{\prime })$ is the single-target Markov transition density, $p_{S}(x^{\prime })$ is the probability that a target with state $x^{\prime }$ at time k−1 will survive at time k, $b_{k|k-1}(x|x^{\prime })$ is the PHD of targets at time k spawned by a single target $x^{\prime }$ at time k−1, and $b_{k|k-1}(x)$ is the PHD of new targets entering the scene at time k.

At time k the sensor collects a new multi-target measurement set $Z_{k}=\left\{z_{1},\cdots ,z_{m}\right\}$, if we assume that the predicted multi-target distribution is approximately Poisson, the closed-form formula of corrector equation of the PHD filter can be derived as
(2)$$D_{k|k}(x)\approx \left[1-p_{D}(x)+p_{D}(x)\sum_{z\in Z_{k}}\frac{L_{z}(x)}{\lambda \cdot c(z)+\int p_{D}(x)L_{z}(x)D_{k|k-1}(x)dx}\right]\cdot D_{k|k-1}(x),$$
where $L_{z}(x)$ is the single-target likelihood function, $p_{D}(x)$ is the probability that a target with state x at time k will be detected, λ is the average number of Poisson-distributed false alarms, the spatial distribution of which is governed by the probability density c(z).

The expected number of targets can be estimated by rounding the integral of the PHD over the entire state space, and then the state-estimates of the targets can be obtained from the local maxima of the PHD.

### 2.2. SMC-PHD Filter

Up to now, PHD filters can be realized by SMC approximation or GM approximation. Compared with the GM-PHD filter, the SMC-PHD filter is suitable for problems involving non-linear non-Gaussian dynamics. Regardless of spawned targets, the following sequentially describes each of the SMC-PHD filter processing steps: initialization, prediction, correction, and state estimation.

Initialization: Suppose prior PHD at time 0 is
(3)$$D_{0|0}(x)\approx \sum_{i=1}^{v_{0|0}}w_{0|0}^{i}\delta (x-x_{0|0}^{i}),$$
where δ(x) is Dirac delta function, $v_{0|0}$ is the number of particles, $x_{0|0}^{i}$ is the *i*th particle and $w_{0|0}^{i}$ is the corresponding weight.

Prediction: Suppose PHD at time k−1 can be approximated using a group of particles
(4)$$D_{k-1|k-1}(x)\approx \sum_{i=1}^{v_{k-1|k-1}}w_{k-1|k-1}^{i}\delta (x-x_{k-1|k-1}^{i}).$$
The meaning of the variables in the above formula is similar to that of Equation (3). Then the predicted PHD at time k is
(5)$$D_{k|k-1}(x)\approx \sum_{i=1}^{v_{k|k-1}}w_{k|k-1}^{i}\delta (x-x_{k|k-1}^{i}),$$
where $v_{k|k-1}=v_{k-1|k-1}+v_{k|k-1}^{birth}$ is the number of predicted particles, $v_{k|k-1}^{birth}$ is the number of appearing particles, $x_{k|k-1}^{i},i=1,\cdots ,v_{k-1|k-1}$ is obtained by the single-target Markov transition density, $x_{k|k-1}^{i},i=v_{k-1|k-1}+1,\cdots ,v_{k|k-1}$ is sampled from the probability density of the spontaneously appearing targets, $w_{k|k-1}^{i}=p_{S}(x_{k-1|k-1}^{i})\cdot w_{k-1|k-1}^{i},i=1,\cdots ,v_{k-1|k-1}$ is the weight corresponding to persisting particles, $w_{k|k-1}^{i}=1/\rho ,i=v_{k-1|k-1}+1,\cdots ,v_{k|k-1}$ is the weight corresponding to appearing particles, and the PHD filter requires ρ particles to adequately maintain track on any individual target.

Correction: After receiving the multi-target measurement set, the posterior PHD at time k can be approximated as
(6)$$D_{k|k}(x)\approx \sum_{i=1}^{v_{k|k}}w_{k|k}^{i}\delta (x-x_{k|k}^{i}),$$
where $v_{k|k}=v_{k|k-1}$ and $x_{k|k}^{i}=x_{k|k-1}^{i},i=1,\cdots ,v_{k|k}$ are the same as those of the predicted PHD, and *i*th particle weight can be calculated by
(7)$$w_{k|k}^{i}=w_{k|k-1}^{i}p_{D}(x_{k|k-1}^{i})\sum_{z\in Z_{k}}\frac{L_{z}(x_{k|k-1}^{i})}{\lambda c(z)+\sum_{e=1}^{v_{k|k-1}}w_{k|k-1}^{e}p_{D}(x_{k|k-1}^{e})L_{z}(x_{k|k-1}^{e})}+w_{k|k-1}^{i}\left[1-p_{D}(x_{k|k-1}^{i})\right].$$
The above particle weights are not equal, and the resampling technique can be utilized to replace them with new, equal weights.

State estimation: The expected number of targets at time k is ${\hat{N}}_{k|k}\approx round\left(\sum_{i=1}^{v_{k|k}}w_{k|k}^{i}\right)$, and the state-estimates of the targets can be obtained by clustering [26,27], data-driven methods [28,29,30,31], and so on.

## 3. Refined PHD Filter

The standard SMC-PHD filter has considered the influence of the detection probability on multi-target tracking. It is indicated from Equation (7) that particle weight of the posterior PHD is a weighted sum of two terms [28,29,30,31]. The first term updates predicted particle weight according to the likelihood function, while the second term directly propagates predicted weight to the posterior PHD considering possible miss detection. Furthermore, the weights of these two terms are detection probability and probability of miss detection, respectively. However, the performance of the standard SMC-PHD filter degrades significantly under low detection probability in practice, especially when continuous miss detection occurs. This is because, continuous miss detection of a target makes the posterior weights small, which may eliminate corresponding particles from the particle pool and then lose the target. This paper proposes a heuristic method called Refined PHD (R-PHD) filter aiming to improve the performance of the SMC-PHD filter under low detection probability. In the proposed method, survival probability dependent on target state is defined, and each target is assigned a label. When miss detection occurs, posterior weights will be revised according to the prediction step. After state estimation of each step, target confirmation is conducted based on sequential probability ratio test.

### 3.1. Survival Probability Dependent on Target State

Measurement of a specific target is not collected due to either miss detection or death of the target. Multi-target tracking algorithms should take some compensation measures for the former reason, while do nothing for the latter. The way to distinguish between the two reasons is to consider the survival probability of the target. If this survival probability is larger than a threshold, algorithms can confirm the target is persisting and compensate miss detection. The earlier versions of the SMC-PHD filter consider the survival probability as a constant which is independent of target state and can’t be used to judge whether the target survives. This paper defines a new survival probability dependent on target state, which is used as one of the conditions to revise posterior particle weights.

Intuitively, targets usually enter sensor FoV from the boundary and exit also from the boundary. The survival probability of a specific target can be very high when it is located in the middle of FoV. On the contrary, the survival probability of a specific target drops rapidly when it is located near the boundary of FoV and moves outwards. Without the loss of generality from an algorithmic viewpoint, this paper considers a rectangular FoV which possesses four boundaries, up and down, left and right, and then the survival probability of a target at time k is
(8)$$p_{S}^{k}=\min\left\{p_{S,up}^{k},p_{S,down}^{k},p_{S,left}^{k},p_{S,right}^{k}\right\},$$
where $p_{S,up}^{k},p_{S,down}^{k},p_{S,left}^{k},p_{S,right}^{k}$ are the survival probabilities of the target with respect to the four boundaries, respectively.

Suppose the particles representing the target at time k are $x_{k|k}^{i},i=1,\cdots ,v_{k|k}$, where each particle is a four-dimensional vector $x_{k|k}^{i}={\left[p_{x}^{i},v_{x}^{i},p_{y}^{i},v_{y}^{i}\right]}^{T}$, representing the target position and velocity along the x-axis and y-axis, respectively, and then the target state and corresponding variance can be estimated as $mean\left(x_{k|k}^{i}\right)$ and $\mathrm{var}\left(x_{k|k}^{i}\right)$. If the particles follow Gaussian distribution and the four variables in particles are independent of each other, the state of this target follows Gaussian distribution $N\left({\left[p_{x},v_{x},p_{y},v_{y}\right]}^{T},diag\left[\sigma_{px}^{2},\sigma_{vx}^{2},\sigma_{py}^{2},\sigma_{vy}^{2}\right]\right)$, where $v_{x}=mean\left(v_{x}^{i}\right)$, $\sigma_{vx}^{2}=\mathrm{var}\left(v_{x}^{i}\right)$ and so on. Based on the above discussion, the survival probability of the target with respect to the right boundary is
(9)$$p_{S,right}^{k}=\mathrm{Pr}\left(u\leq \frac{b_{right}-p_{x}-v_{x}T}{\sqrt{\sigma_{px}^{2}+\sigma_{vx}^{2}T^{2}}}\right),$$
where u~N(0,1), $b_{right}$ is the position of the right boundary, and T is the sampling period. Meanwhile, the survival probabilities of the target with respect to the other three boundaries have similar results.

It should be mentioned that a specific target and corresponding particles share the identical survival probability, which is used for the predictor equation and revising posterior particle weights.

### 3.2. Labeling Target and Particle

In order to confirm if miss detection occurs for each target and identify particles representing the undetected target, every target and particle has its own unique label. On the other hand, the standard SMC-PHD filter can only provide the point-valued estimates of the target states at each time, not track-valued estimates of individual targets due to no record of the target identities. Some principled solutions such as labeled RFS [15,16] were proposed, and produce track-valued estimate without post processing. This paper attaches a unique label to individual targets and particles, which can be used not only for trajectory extraction, but can also compensate miss detection. It should be pointed out that the particles representing a target can have several different labels, and particles with identical labels can also belong to different targets. Labels are assigned to individual targets and individual particles, considering the following principles:

Principles for labeling targets:The label of one target is determined by the label with the largest number of particles belonging to this target.If the label of one target is zero, a new positive number will be assigned to it as its label.When there are multiple targets with the same label at time k, the optimal successor will be selected and keep its label unchanged while others will be assigned a new positive number sequentially.

Principles for labeling particles:4.The label of appearing particles is initialized as zero.5.Particles remain their labels unchanged when surviving.6.The resampling technique doesn’t change the labels of particles.7.If the label of one target is zero, the corresponding particles with label zero will be also assigned a new label corresponding to the target’s new label.8.When there are multiple targets with the same label at time k, the label of the particles representing optimal successor will remain unchanged, while others will be changed with their targets.

It should be mentioned that principle 7 is consistent with principle 2, and principle 8 is consistent with principle 3. False alarm may have the same label as a real target. Consequently, the optimal successor should be selected from all the targets with the same label to inherit the label. Suppose the state of the only target with label l at time k−1 is $x_{l,k-1}$, the states of targets with the same label at time k are $x_{l,k}^{\left(n\right)},n=1,2,\cdots$, then the optimal successor can be selected by comparing the single-target Markov transition density
(10)$$\arg\underset{n}{\max}f_{k|k-1}(x_{l,k}^{\left(n\right)}|x_{l,k-1}),$$

The detailed Algorithm 1 of labelling particles and targets at each time is provided as below:

**Algorithm 1** Labelling Particles and TargetsInitialization: the initialization particles are labelled with zeros, and maximum label is set to r=0.Prediction: labels of the prediction particles are $l_{k|k-1}^{i},i=1,\cdots ,v_{k|k-1}$, where $l_{k|k-1}^{i}=l_{k-1|k-1}^{i},i=1,\cdots ,v_{k-1|k-1}$ and $l_{k|k-1}^{i}=0,i=v_{k-1|k-1}+1,\cdots ,v_{k|k-1}$.Correction: labels of the posterior particles are $l_{k|k}^{i}=l_{k|k-1}^{i},i=1,\cdots ,v_{k|k}$, and the resampling technique doesn’t change the labels of particles.Trajectory extraction: ${\hat{N}}_{k|k}$ targets are extracted from the posterior PHD. The label of target $x_{k}^{\left(n\right)}$ can be determined by $\arg\underset{l}{\max}\left|\left\{l_{k|k}^{i}|l_{k|k}^{i}=l,x_{k|k}^{i}\in x_{k}^{\left(n\right)},i=1,\cdots ,v_{k|k}\right\}\right|,n=1,\cdots ,{\hat{N}}_{k|k}$, where |X| represents the cardinality of set X.  For each target $x_{k}^{\left(n\right)}$, if its label is zero, then r=r+1, set its label to r, and set $l_{k|k}^{i}|l_{k|k}^{i}=0,x_{k|k}^{i}\in x_{k}^{\left(n\right)},i=1,\cdots ,v_{k|k}$ to r.  If there are multiple targets with the same label l, the optimal successor can be selected by Equation (10), and for other target similar operation will be performed like the scene that the label of target is zero.

### 3.3. Revision of Posterior Weights

The posterior particle weights of a specific target will become small if continuous miss detection occurs to it, which may eliminate corresponding particles from the particle pool and then lose the target. In order to maintain the target that is not detected due to low detection probability, this paper replaces the posterior weights with corresponding prediction weights. That is to say, Equation (7) is modified to $w_{k|k}^{i}=w_{k|k-1}^{i}$. However, a target can’t be detected when it disappears from FoV. Therefore, this paper only considers the target whose survival probability is above a threshold $p_{S}^{th}$. Furthermore, only when the sum of posterior weights is less than half of the sum of corresponding prediction weights will this paper conduct revision operations. Revisions of posterior weights are performed after the correction step, and the detailed Algorithm 2 at each time is provided as below:

**Algorithm 2** Revision of Posterior WeightsFor each target $x_{k-1}^{\left(n\right)},n=1,\cdots ,{\hat{N}}_{k-1|k-1}$ at previous time  If the survival probability of $x_{k-1}^{\left(n\right)}$ is above the threshold: $p_{S}^{k-1,(n)}>p_{S}^{th}$, then    Find the prediction weights and posterior weights corresponding to target $x_{k-1}^{\left(n\right)}$:     $w_{k|k-1}^{i},w_{k|k}^{i},i\in I^{\left(n\right)}$, where $I^{\left(n\right)}$ is the set of the index representing target $x_{k-1}^{\left(n\right)}$.    If $sum\left(w_{k|k}^{i}\right)<\frac{1}{2}sum\left(w_{k|k-1}^{i}\right),i\in I^{\left(n\right)}$, do      $w_{k|k}^{i}=w_{k|k-1}^{i},i\in I^{\left(n\right)}$    End  EndEnd

### 3.4. Target Confirmation Based on Sequential Probability Ratio Test

Revisions of posterior weights will bring a new problem: once Poisson-distributed false alarms are captured by the probability density of the spontaneously appearing targets, the proposed algorithm will regard them as targets and maintain corresponding particles and weights by prediction step although no measurement available afterwards. In order to distinguish real targets from false alarms captured by the probability density of newborn targets, the measurement of each target extracted from posterior PHD should be recorded. Suppose $x_{l,k}^{\left(n\right)},n=1,\cdots ,{\hat{N}}_{k|k}$ is the target with label l at time k extracted from the posterior PHD, and the measurement set at time k collected by the sensor is $Z_{k}=\left\{z_{1},\cdots ,z_{m}\right\}$, then the likelihood function of $Z_{k}$ with respect to $x_{l,k}^{\left(n\right)}$ is defined as
(11)$$L_{l,k}=\underset{z\in Z_{k}}{\max}L_{z}(x_{l,k}^{\left(n\right)}).$$
Obviously, the parameter $L_{l,k},k=1,2,\cdots$ can tell us whether the target with label l is a real target. For simplification, the proposed algorithm binarizes the above likelihood function as
(12)$${L^{\prime }}_{l,k}=\{\begin{array}{c} 1,L_{l,k}\geq L^{th}\\ 0,L_{l,k}<L^{th} \end{array},$$
where $L^{th}$ is the threshold judging if there is a measurement of one target. The probability that there exists corresponding measurement of target $x_{l,k}^{\left(n\right)}$ is
(13)$$\begin{array}{l} \mathrm{Pr}\left({L^{\prime }}_{l,k}=1\right)\\ =\mathrm{Pr}\left(\underset{z\in Z_{k}}{\max}L_{z}(x_{l,k}^{\left(n\right)})\geq L^{th}\right)\\ =1-\mathrm{Pr}\left(L_{z}(x_{l,k}^{\left(n\right)})<L^{th},\forall z\in Z_{k}\right) \end{array}$$
Furthermore, the bigger the cumulative sum $\underset{k}{sum}{L^{\prime }}_{l,k}$, the more we can confirm that the target with label l is a real target.

In order to confirm targets in real time, this paper proposes the method based on sequential probability ratio test. Without loss of generality, suppose the single-target likelihood function is Gaussian
(14)$$L_{z}(x)=\sqrt{\frac{1}{{\left(2\pi \right)}^{2}\det(C)}}\exp\left(-\frac{1}{2}{\left(z-H\left(x\right)\right)}^{T}C^{-1}\left(z-H\left(x\right)\right)\right),$$
where $z={\left[z_{1},z_{2}\right]}^{T}$, $C=diag\left[\sigma_{1}^{2},\sigma_{2}^{2}\right]$ is the covariance matrix of observation noises, H(x) is the deterministic state-to-measurement transform model, and target x is located at the coordinate origin, the probability that the likelihood function of single measurement is above the threshold is
(15)$$\mathrm{Pr}\left(L_{z}(x)>L^{th}\right)=\mathrm{Pr}\left(\frac{z_{1}^{2}}{\sigma_{1}^{2}}+\frac{z_{2}^{2}}{\sigma_{2}^{2}}<-2\ln\left(2\pi L^{th}\sqrt{\sigma_{1}^{2}\sigma_{2}^{2}}\right)\right)=\iint_{\frac{z_{1}^{2}}{\sigma_{1}^{2}}+\frac{z_{2}^{2}}{\sigma_{2}^{2}}<-2\ln\left(2\pi L^{th}\sqrt{\sigma_{1}^{2}\sigma_{2}^{2}}\right)}f(z_{1},z_{2})dz_{1}dz_{2},$$
which indicates the measurement z lies inside the ellipse $z_{1}^{2}/\sigma_{1}^{2}+z_{2}^{2}/\sigma_{2}^{2}=-2\ln\left(2\pi L^{th}\sqrt{\sigma_{1}^{2}\sigma_{2}^{2}}\right)$ and where $f(z_{1},z_{2})$ is the spatial distribution of z. Suppose false alarms obey uniform distribution spatially, then the probability that the likelihood function of single clutter is above the threshold is $p_{0}=S_{e}/S_{FOV}$, where $S_{e}$ is the area of the above ellipse and $S_{FOV}$ is the area of the whole FoV. If a real target is detected, the probability that the likelihood function of target measurement is above the threshold is $q_{0}=\iint_{\frac{z_{1}^{2}}{\sigma_{1}^{2}}+\frac{z_{2}^{2}}{\sigma_{2}^{2}}<-2\ln\left(2\pi L^{th}\sqrt{\sigma_{1}^{2}\sigma_{2}^{2}}\right)}L_{z}(x)dz_{1}dz_{2}$. Suppose $\sigma^{2}=\sigma_{1}^{2}=\sigma_{2}^{2}$, then the probability $q_{0}$ with respect to observation noise variance $\sigma^{2}$ under different thresholds is depicted in Figure 1, which indicates that $q_{0}$ is close to 1 under a suitable threshold.

Next, we consider the situation of multiple measurements. When the target $x_{l,k}^{\left(n\right)}$ is a false alarm, the measurement set $Z_{k}$ can be organized as the union of measurements from targets and clutter. The likelihood function of $Z_{k}$ with respect to $x_{l,k}^{\left(n\right)}$ is
(16)$$L_{l,k}=\max\left\{\underset{z\in Z_{k}\backslash K_{k}}{\max}L_{z}(x_{l,k}^{\left(n\right)}),\underset{z\in K_{k}}{\max}L_{z}(x_{l,k}^{\left(n\right)})\right\},$$
Therefore, Equation (13) is
(17)$$\begin{array}{l} \mathrm{Pr}\left({L^{\prime }}_{l,k}=1\right)\\ =1-\mathrm{Pr}\left(L_{z}(x_{l,k}^{\left(n\right)})<L^{th},\forall z\in Z_{k}\backslash K_{k}\right)\mathrm{Pr}\left(L_{z}(x_{l,k}^{\left(n\right)})<L^{th},\forall z\in K_{k}\right)\\ \approx 1-\mathrm{Pr}\left(L_{z}(x_{l,k}^{\left(n\right)})<L^{th},\forall z\in K_{k}\right)\\ =1-{\left(\mathrm{Pr}\left(L_{z}(x_{l,k}^{\left(n\right)})<L^{th},z\in K_{k}\right)\right)}^{D}\\ =1-{\left(1-p_{0}\right)}^{D} \end{array}$$
where D is the number of clutters, following $\mathrm{Pr}\left(D=d\right)=\lambda^{d}e^{-\lambda }/d!,d=0,1,\cdots$. The approximation is reasonable, because the false alarm always appears before or after corresponding real target, which results that it can neither be associated with the measurement of its corresponding real target nor that of other real targets. Considering D is a random variable, the expectation of Equation (17) is
(18)$$E\left[1-{\left(1-p_{0}\right)}^{D}\right]=1-\sum_{d=0}^{\infty }{\left(1-p_{0}\right)}^{d}\frac{\lambda^{d}e^{-\lambda }}{d!}=1-e^{-p_{0}\lambda }.$$

On the other hand, when the target $x_{l,k}^{\left(n\right)}$ is a real target, the measurement set $Z_{k}$ can be divided into three parts: the measurement generated from target $x_{l,k}^{\left(n\right)}$, measurements generated from other targets and clutters. The likelihood function of $Z_{k}$ with respect to $x_{l,k}^{\left(n\right)}$ is
(19)$$L_{l,k}=\max\left\{L_{\Theta_{k}(x_{l,k}^{\left(n\right)})}(x_{l,k}^{\left(n\right)}),\underset{z\in K_{k}}{\max}L_{z}(x_{l,k}^{\left(n\right)}),\underset{z\in Z_{k}\backslash K_{k}\backslash \Theta_{k}(x_{l,k}^{\left(n\right)})}{\max}L_{z}(x_{l,k}^{\left(n\right)})\right\}.$$
Therefore, Equation (13) is
(20)$$\begin{array}{l} \mathrm{Pr}\left({L^{\prime }}_{l,k}=1\right)\\ =p_{D}(x_{l,k}^{\left(n\right)})\left[1-\left(1-q_{0}\right){\left(1-p_{0}\right)}^{D}\mathrm{Pr}\left(L_{z}(x_{l,k}^{\left(n\right)})<L^{th},\forall z\in Z_{k}\backslash K_{k}\backslash \Theta_{k}(x_{l,k}^{\left(n\right)})\right)\right]\\ +\left(1-p_{D}(x_{l,k}^{\left(n\right)})\right)\left[1-{\left(1-p_{0}\right)}^{D}\mathrm{Pr}\left(L_{z}(x_{l,k}^{\left(n\right)})<L^{th},\forall z\in Z_{k}\backslash K_{k}\right)\right]\\ \approx p_{D}(x_{l,k}^{\left(n\right)})\left[1-\left(1-q_{0}\right){\left(1-p_{0}\right)}^{D}\right]+\left(1-p_{D}(x_{l,k}^{\left(n\right)})\right)\left[1-{\left(1-p_{0}\right)}^{D}\right]\\ \approx p_{D}(x_{l,k}^{\left(n\right)})+\left(1-p_{D}(x_{l,k}^{\left(n\right)})\right)\left[1-{\left(1-p_{0}\right)}^{D}\right]\\ \approx p_{D}(x_{l,k}^{\left(n\right)}) \end{array}$$
in which we consider if the real target $x_{l,k}^{\left(n\right)}$ is detected. Three approximations are reasonable when all real targets are far from each other, $q_{0}$ is close to 1, and $p_{D}(x_{l,k}^{\left(n\right)})>>1-e^{-p_{0}\lambda }$, respectively.

In summary, random variable ${L^{\prime }}_{l,k}$ obeys two-point distribution
(21)$$\mathrm{Pr}\left({L^{\prime }}_{l,k}=1\right)=p,\mathrm{Pr}\left({L^{\prime }}_{l,k}=0\right)=1-p,$$
where success probability $p=p_{1}=1-e^{-p_{0}\lambda }$ when the target with label l is a false alarm, and $p=p_{2}=p_{D}(x_{l,k}^{\left(n\right)})$ when the target with label l is a real target. Then, target confirmation can be represented as a hypothesis test problem
(22)$$H:p=p_{1}↔K:p=p_{2}.$$
SPRT tells us: when $\underset{k}{sum}{L^{\prime }}_{l,k}\leq A_{n}$ is true, to accept H, mark $x_{l,k}^{\left(n\right)}$ as a false alarm and eliminate corresponding particles; when $\underset{k}{sum}{L^{\prime }}_{l,k}\geq B_{n}$ is true, to reject H and mark $x_{l,k}^{\left(n\right)}$ as a real target; otherwise, to make no decision and maintain particles of $x_{l,k}^{\left(n\right)}$. The parameters $A_{n}$ and $B_{n}$ are
(23)$$\begin{array}{l} A_{n}=\left(\frac{\beta }{1-\alpha }-n\ln\frac{1-p_{2}}{1-p_{1}}\right)/\ln\frac{p_{2}\left(1-p_{1}\right)}{p_{1}\left(1-p_{2}\right)}\\ B_{n}=\left(\frac{1-\beta }{\alpha }-n\ln\frac{1-p_{2}}{1-p_{1}}\right)/\ln\frac{p_{2}\left(1-p_{1}\right)}{p_{1}\left(1-p_{2}\right)} \end{array}$$
where α, β are Type Ⅰ error rate and Type Ⅱ error rate, respectively, and n is the cumulative time of the target with label l from emerging to current step.

It should be mentioned that confirmation of a real target always lags behind its emerging. Fortunately, we can make up point-valued estimates of the target at previous times.

### 3.5. Refined PHD Filter

The key modules of the refined PHD filter were explained in the previous subsections. Here, we summarize the overall steps of the proposed method in Algorithm 3.

**Algorithm 3** Refined PHD FilterInitialization: suppose prior PHD at time 0 is Equation (3), $l_{0|0}^{i}=0,i=1,\cdots ,v_{0|0}$, and r=0.Prediction: the predicted PHD at time k is Equation (5), and labels of the prediction particles are $l_{k|k-1}^{i}=l_{k-1|k-1}^{i},i=1,\cdots ,v_{k-1|k-1}$ and $l_{k|k-1}^{i}=0,i=v_{k-1|k-1}+1,\cdots ,v_{k|k-1}$.Correction: the posterior particle weights at time k are calculated by Equation (7), and $l_{k|k}^{i}=l_{k|k-1}^{i},i=1,\cdots ,v_{k|k}$.Revision of Posterior Weights: execute revision of posterior weights introduced in Section 3.3, the revised weights are still represented as $w_{k|k}^{i}$, and the posterior PHD at time k is Equation (6).State estimation: ${\hat{N}}_{k|k}\approx round\left(\sum_{i=1}^{v_{k|k}}w_{k|k}^{i}\right)$, resample with no change of particle labels, and ${\hat{N}}_{k|k}$ targets are extracted by k-means clustering: $x_{k}^{\left(n\right)},n=1,\cdots ,{\hat{N}}_{k|k}$.Trajectory extraction: determine the label of target $x_{k}^{\left(n\right)}$ and corresponding particles according to Section 3.2.Survival Probability Calculation: calculate survival probability of individual target according to Section 3.1.Target Confirmation: calculate test statistic $\underset{k}{sum}{L^{\prime }}_{l,k}$, and for individual target, add it to the confirmation set, discard it, or make no decision according to Section 3.4.

## 4. Simulation

### 4.1. Simulation Scenery

In this section, we use computer simulations to demonstrate the effectiveness and performance of the proposed method. Suppose FoV is a two-dimensional region [−50,50]×[0,100] in which multiple targets appear or disappear at any time. The state equation and the measurement equation of single target can be represented as follows:(24)$$x_{k}=\left[\begin{array}{cccc} 1 & T & 0 & 0\\ 0 & 1 & 0 & 0\\ 0 & 0 & 1 & T\\ 0 & 0 & 0 & 1 \end{array}\right]x_{k-1}+\left[\begin{array}{cc} T^{2}/2 & 0\\ T & 0\\ 0 & T^{2}/2\\ 0 & T \end{array}\right]\left[\begin{array}{c} n_{1}\\ n_{2} \end{array}\right],$$
(25)$$z_{k}=\left[\begin{array}{cccc} 1 & 0 & 0 & 0\\ 0 & 0 & 1 & 0 \end{array}\right]x_{k}+\left[\begin{array}{c} w_{1}\\ w_{2} \end{array}\right],$$
where target state $x_{k}={\left[p_{xk},v_{xk},p_{yk},v_{yk}\right]}^{T}$ consists of the target position and velocity along the x-axis and y-axis, only target position is measured represented as $z_{k}$, sampling period T=1, and the process noise and the measurement noise are both zero mean Gaussian noises: ${\left[n_{1},n_{2}\right]}^{T}\sim N\left({\left[0,0\right]}^{T},diag\left[0.01,0.01\right]\right)$, ${\left[w_{1},w_{2}\right]}^{T}\sim N\left({\left[0,0\right]}^{T},diag\left[0.09,0.09\right]\right)$. This paper considers five targets with motion parameters showed in Table 1, and the total time of simulation is $T_{total}=100$. Figure 2 depicts the simulation scenery in x-y coordinate system.

Cardinality and Optimal Sub-Pattern Assignment (OSPA) distance [43] between real set of target states and estimated set of target states are employed as performance evaluation criterions, and the cut-off factor and the order used in OSPA are c=10, p=2, respectively. The performance of the proposed Refined PHD (R-PHD) filter is evaluated in comparison with the standard PHD filter, CPHD filter, and CBMeMBer filter, and the filters here are all implemented with SMC implementations. Survival probability is set as 0.99 in PHD, CPHD and CBMeMBer. In R-PHD, the threshold $L^{th}$ and $p_{S}^{th}$ are set as 0.1 and 0.5 respectively, and Type Ⅰ and Ⅱ error rates are set as α=0.1 and β=0.1, respectively. In all four filters, 1000 particles are used for per target, and the probability density of newborn targets is modeled as Gaussian mixture of target initial states with the covariance of diag[1,0.1,1,0.1].

### 4.2. Evaluation of Different Detection Probabilities

Figure 3 depicts the mean OSPA and cardinality versus time over P=200 Monte-Carlo runs, where the detection probability is set as $p_{D}=0.85$, independent of target state, and the average number of Poisson-distributed false alarms is set as λ=10. Mean OSPA is depicted in Figure 3a, and each data point is calculated as
(26)$$\frac{1}{P}\sum_{p=1}^{P}OSPA_{p,k},$$
where $OSPA_{p,k}$ is OSPA distance at time k in *p*th Monte-Carlo trial. Mean cardinality is depicted in Figure 3b, and each data point is calculated as
(27)$$\frac{1}{P}\sum_{p=1}^{P}{\hat{N}}_{p,k|k},$$
where ${\hat{N}}_{p,k|k}$ is the estimated number of targets at time k in *p*th Monte-Carlo trial. Due to low detection probability, the measurements from targets are intermittent, and the PHD filter, CPHD filter and CBMeMBer filter can’t obtain excellent results. The mean OSPA of the R-PHD filter is usually smaller than that of the competing methods, and the mean cardinality of the R-PHD filter is closer to the ground truth. It is worth noting that OSPA distances at time 11 and 91 are apparently large in all filters, due to simultaneous birth or death of two targets. Figure 3 illustrates that the proposed method can effectively track multiple targets under low detection probability.

Then, we compare multi-target tracking performances of different methods with respect to different detection probabilities from $p_{D}=0.7$ to $p_{D}=1$. Figure 4 illustrates the multi-target tracking results, where the average number of false alarms is set as λ=10 for all simulations. Mean OSPA with respect to different detection probabilities is depicted in Figure 4a, and each data point is calculated as
(28)$$\frac{1}{PT_{total}}\sum_{p=1}^{P}\sum_{k=1}^{T_{total}}OSPA_{p,k},$$
where $OSPA_{p,k}$ is OSPA distance at time k in *p*th Monte-Carlo trial. Mean Root Mean Square Error (RMSE) of cardinality with respect to different detection probabilities is depicted in Figure 4b, and each data point is calculated as
(29)$$\frac{1}{T_{total}}\sum_{k=1}^{T_{total}}\sqrt{\frac{1}{P}\sum_{p=1}^{P}{\left(N_{k}-{\hat{N}}_{p,k|k}\right)}^{2}},$$
where ${\hat{N}}_{p,k|k}$ is the estimated number of targets at time k in *p*th Monte-Carlo trial, and $N_{k}$ is real number of targets at time k. Figure 4 shows that mean OSPA and mean RMSE of cardinality both decrease monotonically as detection probability increases in the PHD filter, CPHD filter and CBMeMBer filter. In addition, the performance of the R-PHD filter is relatively stable, and the proposed method presents better performance than baselines when detection probability is no more than 0.95. It should be mentioned that when detection probability is 1, that is to say, there is no miss detection, the R-PHD filter has inferior performance than the other three methods, which can be explained by the Type Ⅰ error rate in SPRT.

### 4.3. Evaluation of Different Average Numbers of False Alarms

Figure 5 depicts the mean OSPA and cardinality versus time over P=200 Monte-Carlo runs, where the detection probability is set as $p_{D}=0.95$, independent of target state, and the average number of Poisson-distributed false alarms is set as λ=20. Mean OSPA is depicted in Figure 5a, and each data point is calculated using Equation (26). Mean cardinality is depicted in Figure 5b, and each data point is calculated using Equation (27). Showed in Figure 5a, the mean OSPA of the R-PHD filter is usually smaller than that of the PHD filter and CBMeMBer filter, while it is bigger than that of the CPHD filter at most steps. Figure 5b illustrates that the number of targets estimation of the CPHD filter always lags behind the ground truth when target birth or target death occurs.

Figure 6 illustrates multi-target tracking performances of different methods with respect to different average numbers of false alarms from λ=10 to λ=30, where the detection probability is set as $p_{D}=0.95$. Mean OSPA with respect to different average numbers of false alarms is depicted in Figure 6a, and each data point is calculated using Equation (28). Mean RMSE of cardinality with respect to different average numbers of false alarms is depicted in Figure 6b, and each data point is calculated using Equation (29). Figure 6 shows that multi-target tracking performances of different methods deteriorate slightly as average number of false alarms increases. Furthermore, the R-PHD filter outperforms the other three methods at λ=10, while it has inferior OSPA performance compared to the PHD filter and CPHD filter and inferior cardinality performance compared to the CPHD filter at λ=30. That is because the hypothesis $p_{D}>>1-e^{-p_{0}\lambda }$ is no longer valid when the number of clutters is considerable. Generally, the proposed method can provide a satisfactory result under high average numbers of false alarms.

### 4.4. Evaluation of Different Continuous Miss detection Durations

Next, we consider the scenario that targets are undetected for continuous steps. Figure 7 shows the multi-target tracking results of different methods under continuous miss detection during 45≤k≤51, detection probability in other steps is set as $p_{D}=0.9$ and average number of false alarms is set as λ=10 in simulations. Mean OSPA in Figure 7a is obtained by averaging 200 trials of Monte-Carlo simulation using Equation (26), and mean cardinality in Figure 7b is obtained using Equation (27). Figure 7 demonstrates that the PHD filter and CBMeMBer filter lose all targets when continuous miss detection during 45≤k≤51 occurs, which results that mean OSPA is up to the cut-off factor and mean cardinality is close to 0 from k=45 to k=100. The CPHD filter loses four targets when continuous miss detection occurs, while it can maintain one target after k=51. Evidently, the proposed R-PHD filter can maintain all targets and its performance is almost immune to continuous miss detection.

Figure 8 illustrates multi-target tracking performances of different methods with respect to different continuous miss detection durations from s=3 to s=11, where the detection probability is set as $p_{D}=0.9$, average number of false alarms is set as λ=10. Continuous miss detection duration is represented as s, and it always begins at time k=45. That is to say, s=3 indicates that targets are missed during 45≤k≤47. The mean OSPA with respect to different continuous miss detection durations is depicted in Figure 8a, and each data point is calculated using Equation (28). Mean RMSE of cardinality with respect to different continuous miss detection durations is depicted in Figure 8b, and each data point is calculated using Equation (29). The performances of the PHD filter, CPHD filter and CBMeMBer filter deteriorate as continuous miss detection duration increases, while that of the proposed R-PHD filter is relatively stable and always superior than the other three methods. In conclusion, the proposed R-PHD filter can effectively track multiple targets when continuous miss detection occurs.

## 5. Conclusions

In this paper, a heuristic method called the refined PHD filter is proposed to improve the multi-target tracking performance of the PHD filter under low detection probability in practice. First, survival probability dependent on target state is defined, which is one of the conditions of performing posterior weights revision. Then, we label individual targets and particles, which can be utilized to confirm if miss detection occurs for each target and identify particles representing the undetected target. In addition, it can provide track-valued estimates of individual targets. When miss detection occurs due to low detection probability, posterior particle weights will be revised according to the prediction step. In order to distinguish real targets and false alarms in real time, we regard the target confirmation problem as a hypothesis test problem and introduce sequential probability ratio test to judge the success probability of the two-point distribution. Simulation results with respect to various detection probabilities, average numbers of false alarms and continuous miss detection durations are provided, which indicates that the multi-target tracking performance of the R-PHD filter outperforms the competing methods.

## Figures and Tables

**Figure 1 sensors-19-02842-f001:**
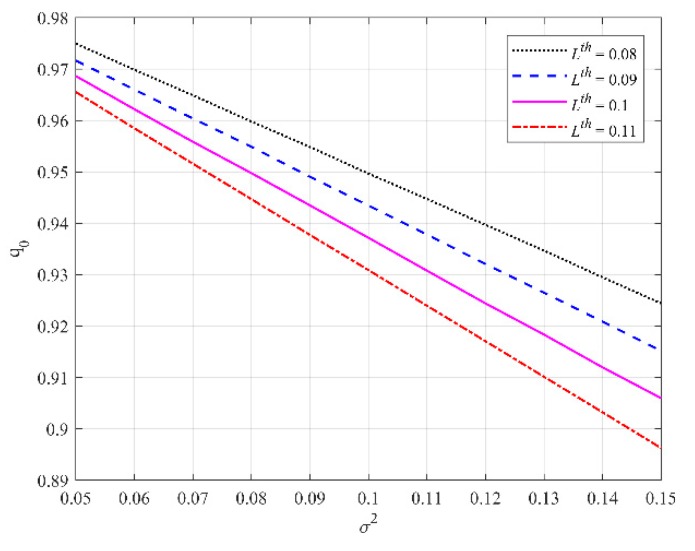
Probability that the likelihood function of target measurement is above the threshold.

**Figure 2 sensors-19-02842-f002:**
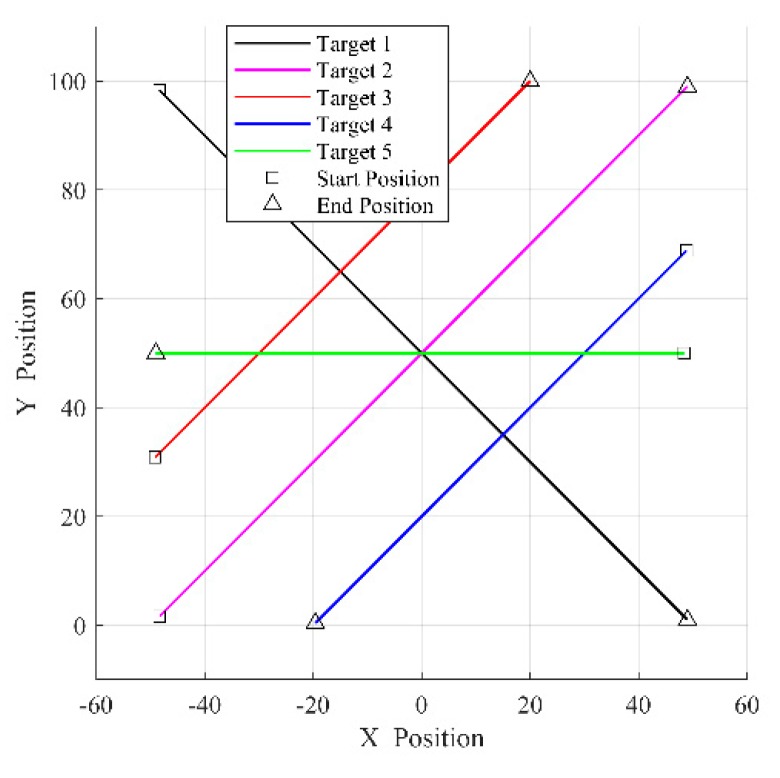
Simulation scenery in x-y coordinate system.

**Figure 3 sensors-19-02842-f003:**
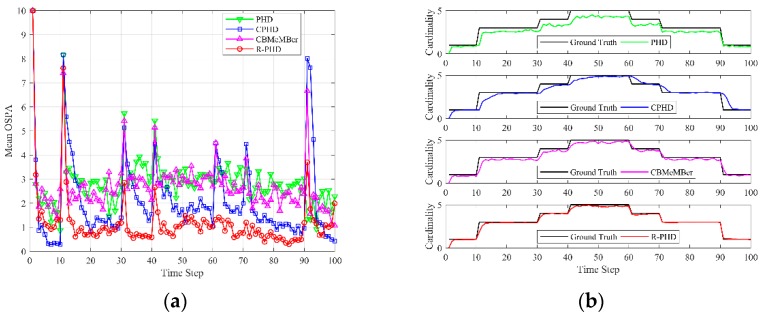
OSPA and cardinality performances of different methods versus time ($p_{D}=0.85$, λ=10): (**a**) mean OSPA; (**b**) mean cardinality.

**Figure 4 sensors-19-02842-f004:**
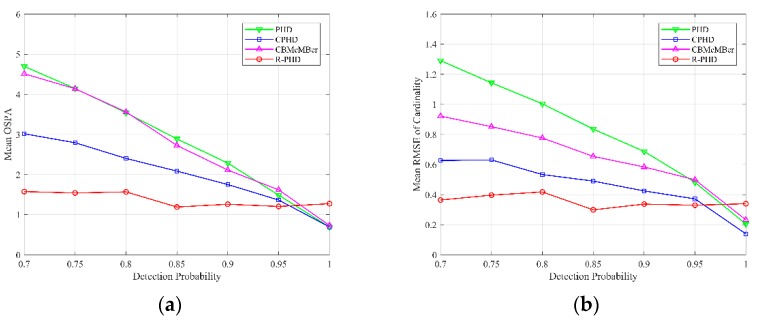
OSPA and cardinality performances of different methods with respect to different detection probabilities from $p_{D}=0.7$ to $p_{D}=1$ (λ=10): (**a**) mean OSPA; (**b**) mean RMSE of cardinality.

**Figure 5 sensors-19-02842-f005:**
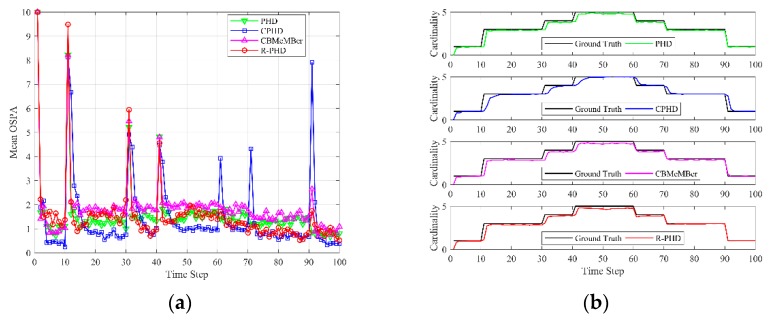
OSPA and cardinality performances of different methods versus time ($p_{D}=0.95$, λ=20): (**a**) mean OSPA; (**b**) mean cardinality.

**Figure 6 sensors-19-02842-f006:**
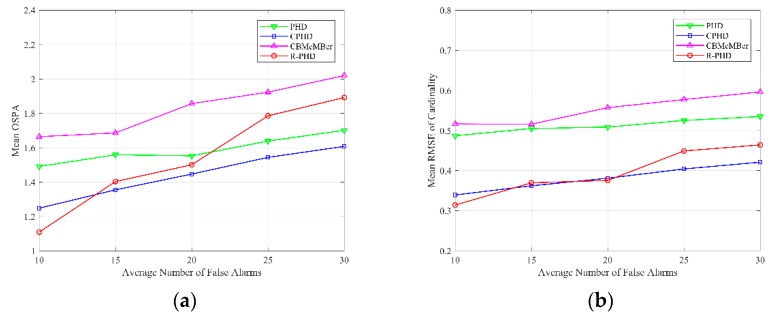
OSPA and cardinality performances of different methods with respect to different average numbers of false alarms from λ=10 to λ=30 ($p_{D}=0.95$): (**a**) mean OSPA; (**b**) mean RMSE of cardinality.

**Figure 7 sensors-19-02842-f007:**
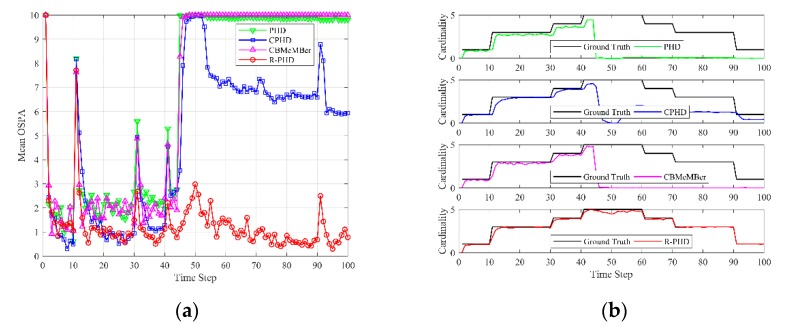
OSPA and cardinality performances of different methods versus time under continuous miss detection during 45≤k≤51 ($p_{D}=0.9$, λ=10): (**a**) mean OSPA; (**b**) mean cardinality.

**Figure 8 sensors-19-02842-f008:**
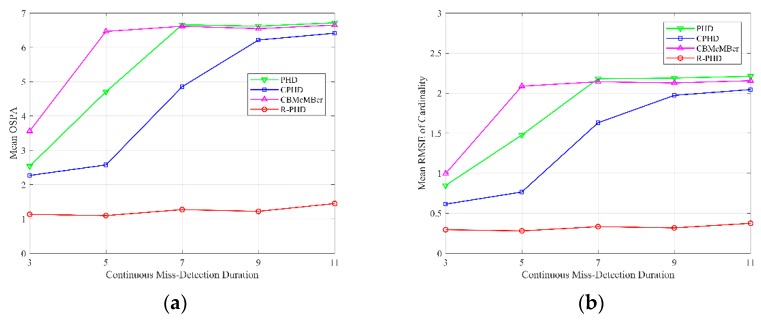
OSPA and cardinality performances of different methods with respect to different continuous miss detection durations from s=3 to s=11 ($p_{D}=0.9$, λ=10): (**a**) mean OSPA; (**b**) mean RMSE of cardinality.

**Table 1 sensors-19-02842-t001:** Motion parameters of targets

Target	Initial State	Birth Time	Death Time
1	${\left[-50,1.65,100,-1.65\right]}^{T}$	1	60
2	${\left[-50,1.65,0,1.65\right]}^{T}$	11	70
3	${\left[-50,0.875,30,0.875\right]}^{T}$	11	90
4	${\left[50,-1.16,70,-1.16\right]}^{T}$	31	90
5	${\left[50,-1.65,50,0\right]}^{T}$	41	100

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
