# Peer review of "Refined PHD Filter for Multi-Target Tracking under Low Detection Probability"

_sensors, 2019, doi:10.3390/s19132842_

Round 1
Reviewer 1 Report
The paper proposes a technique to improve the performance of the PHD filter when detection probability is low (relatively). This is an important problem because the PHD filter by assumption requires a high detection probability, but in practice, especially for radars, the detection probability is usually lower. Hence, to be of practical value the PHD filter must be amended to work with lower detection probability. The proposed technique modifies the survival probability to prevent terminating tracks prematurely. Moreover, addition processing is proposed to modify the updated particles weights to cope with misdetections. The authors also use a hypothesis testing technique to distinguish between target and clutter to improve estimation. The proposed solution has been compared to the standard PHD, CPHD and multi-Bernoulli filter to show improvements.
Overall the paper is well written, the technical content is sound and the numerical studies are convincing. The authors are very forthcoming to state that their method is heuristic, which the reviewer really appreciates. There is nothing wrong with heuristic methods as long as they work, which is what the authors have shown. In light of these observations, I believe the paper should be published.
However, there are some minor comments which should be addressed:
In the abstract and introduction (line 70-71 page 2) the authors state "Standard multi-target tracking algorithms based on Random Finite Sets (RFS) have considered the influence of the detection probability on multi-target tracking, but their performances degrade significantly under low detection probability in practice". This is quite accurate
(a) In low detection probability the performance of all trackers will degrade not just RFS filters.
(b) Due to the approximation in unlabled RFS filters, their performance degrade rapidly with decreasing detection probability. However, labeled RFS filter such as “The Multi-Scan Generalized Labeled Multi-Bernoulli Filter”, produces very good tracking results with detection probability of as low as 0.66, see also demos at the like https://arxiv.org/abs/1805.10038.
The abstract/introduction should be amended to provide a more accurate picture of whats going on as well as updated with recent RFS filters rather than the PHD/CPHD filter (which are quite out dated).
2. 1st sentence of introduction "The objective of Multi-Target Tracking (MTT) is to jointly estimate the number of targets and 32 their individual state..." This is not quite accurate, in tracking one needs to need to also provide target tracks or trajectories.
3.1st paragraph of page 2, ".... multi-target states and sensor measurements as RFS, which avoids data association problem". This is no longer true, recent RFS filters such as the labeled RFS filters (see e.g. [21, 22]) also solve the data association problems. The discussion needs to be modified to include RFS filters which solves the data association problem.
4. 2nd paragraph of page 2, "The multi-target Bayes filter is difficult to implement due to complex combinatorial logic, large computational complexity and large storage complexity". This is no longer true, “an efficient implementation of the generalized labeled multi-Bernoulli filter” published in 2017 has linear complexity in the number of measurements and can handle over a million targets simultaneously, see the demo at the link https://arxiv.org/abs/1804.06622. There are also work on “Multiple object tracking in unknown backgrounds with labeled random finite sets” that can handle non-constant PD, as well as unknown PD. The discussion should be amended to reflect the latest development in RFS filters and not just restricted to unlabeled RFS.
5. The authors employ a heuristic labelling technique to generate tracks. A discussion of labelling techniques that includes principled solution such as labeled RFS [21, 22] should be included to provide some context to the authors contributions.
6. The author uses the OSPA distance a lot on their numerical section but does not provide any references for this distance.
7. It would be interesting to see what the performance is like with a PD of 0.66 as in the multi-scan GLMB filter.
Reviewer 2 Report
The work proposes a heuristic method called Refined PHD (R-PHD) filter to improve multi-target tracking performance under low detection probability. The work is well motivated as it is an important issue to address the low detection probability for multi-target tracking. The key idea is technically sound and innovative. I am positive with the acceptance of the work for publication after some revisions as addressed below.
1, The introduction about the CBMeMBer, GLMB and LMB filters as given in lines 62-69 which are very different from the PHD/CPHD filters is unnecessary and can be reduced. The first two paragraphs of the Introduction section regarding the non-RFS filters can also be reduced to save space. The introduction should be focused on the PHD filter, the theme of the paper, to avoid distracting the readers. The reference should be so too. Actually, as shown below, there are many relevant references on the PHD filter missed.
Following the same line of thinking, section 2.1 can be removed, which is completely well-known knowledge.
2. In lines 88-90 of page 2, it was addressed that “survival probability dependent on target state is defined, which is based on the hypothesis that target entering and exiting sensor Fields of View (FoV) usually occur at the boundary.” ---here, should “target entering and” be “an”? ---- This thinking has been earlier proposed and used in designing the new-target birth model in the following work (and the reference therein):
[*]T. Li, V. Elvira, H. Fan, J.M. Corchado. Local-Diffusion-based Distributed SMC-PHD Filtering Using Sensors with Limited Sensing Range, IEEE Sensors Journal, vol.19, no.4, 2019, pp. 1580 – 1589.
3. Regarding the state estimation step of the filter as given in lines 180-182 of page 5, there have actually been many better solutions than clustering, which are more accurate and computationally faster, such as the data-driven methods [a1-a4]. The authors may consider them for replacement.
[a1] L. Zhao, P. Ma, X. Su, and H. Zhang, “A new multi-target state estimation algorithm for PHD particle filter,” in Proc. FUSION 2010, Edinburgh, Scotland, UK, Jul. 2010.
[a2] T. Li, J. M. Corchado, S. Sun and H. Fan, Multi-EAP: extended EAP for multi-estimate extraction for the SMC-PHD filter, Chinese Journal of Aeronautics, 30(1): 368–379, 2017.
[a3] M. Schikora, W. Koch, R. Streit, and D. Cremers, “Sequential Monte Carlo method for multi-target tracking with the intensity filter,” in Advances in Intelligent Signal Processing and Data Mining: Theory and Applications, P. Georgieva, L. Mihaylova, and L. C. Jain, Eds. Heidelberg, Germany: Springer, 2012, ch. 4, pp. 55–87.
[a4] T. Li, S. Sun, M. Bolic, J. M. Corchado, Algorithm design for parallel implementation of the SMC-PHD filter, Signal Processing, vol.119, pp. 115-127, 2016.
4. As addressed in lines 185-188, the proposed filter is based on the phenomenon that “particle weight of the posterior PHD is a weighted sum of two terms.” This has been observed by the above-mentioned references [a1-a4] and has motivated the estimate extraction algorithms. The authors should mention this fact.
5. The principles for labeling targets and particles are analogous to that of the so-called particle-dyeing approach (A particle dyeing approach for track continuity in the SMC-PHD filter, FUSION’14); please check and address the difference or similarity. To the reviewer, both algorithms are essentially a kind of soft decision. As long as I know, this soft decision works well in case of well distant targets (as demonstrated in the simulation) but may have problems when targets move close to each other or cross over each other.
